# The Trends in Opioid Use in Castile and Leon, Spain: A Population-Based Registry Analysis of Dispensations in 2015 to 2018

**DOI:** 10.3390/jcm8122148

**Published:** 2019-12-05

**Authors:** Francisco Herrera-Gómez, Eduardo Gutierrez-Abejón, Ignacio Ayestarán, Paloma Criado-Espegel, F. Javier Álvarez

**Affiliations:** 1Pharmacological Big Data Laboratory, Pharmacology and Therapeutics, Faculty of Medicine, University of Valladolid, 47005 Valladolid, Spain; egutierreza@saludcastillayleon.es (E.G.-A.); ija@oftalvision.es (I.A.); alvarez@med.uva.es (F.J.Á.); 2Nephrology, Hospital Virgen de la Concha—Sanidad de Castilla y León, 49022 Zamora, Spain; 3Technical Direction of Pharmaceutical Assistance, Gerencia Regional de Salud de Castilla y León, 47007 Valladolid, Spain; 4Gerencia de Asistencia Sanitaria-Sanidad de Castilla y León, 34001 Palencia, Spain; pcriado@saludcastillayleon.es; 5CEIm, Hospital Clínico Universitario de Valladolid—Sanidad de Castilla y León, 47003 Valladolid, Spain

**Keywords:** analgesics, opioids, drug use, driving under the influence, population surveillance

## Abstract

Opioids are driving-impairing medicines (DIM). To assess the evolution and trends of opioid analgesics use between 2015 and 2018 in Castile and Leon (Spain), a population-based registry study was conceived. The length of opioid use and its concomitant use with other DIMs were studied. Analyses were done considering age and gender distributions. Adjusted consumption for licensed drivers is also presented. Of the 5 million dispensations recorded between 2015 and 2018, opioid analgesics were dispensed to 11.44% of the general population and 8.72% of vehicle drivers. Increases among daily users (2.6 times higher) and chronic users (1.5% higher) were noted, supporting the overall increase in opioid use (1.5%). The use of multiple drugs including other DIMs was a common finding (mean ± SD, 2.54 ± 0.01). Acute use (5.26%) and chronic use (3.20%) were also frequent. Formulations combining opioid analgesics with nonopioid analgesics were preferred. The use of opioids increased in Spain between 2015 and 2018. Concomitant use with other DIMS especially affects women and the elderly. Frequent use of opioid analgesics with other DIMs is a serious problem for drivers and increases the risk of accidents. Promoting safe driving should be a main objective of health authorities, to be achieved by developing and implementing educational activities for healthcare professionals and patients.

## 1. Introduction

The nontherapeutic use of opioid analgesics has increased in the last 15 years, to the point where it is now perceived as a serious public health concern [1]. An increase in the number of cases of misuse, abuse, and death in most developed countries has been noted and attributed to the increase in opioid prescriptions [2], even considering disparities between world regions (the medical and nonmedical use of opioids have increased to a lower degree in Europe and Australia compared to Canada and the USA [3,4,5]). Whatever way we look at it, the problem is real: in Canada, between 2016 and 2017, accidental deaths due to the misuse of fentanyl or fentanyl analogues increased by 81% [2]; in the USA, in 2017, 11.4 million people aged 12 years or older misused prescription opioids and 47,600 people died from opioid overdose [6]. Clinicians, researchers, policy-makers, and drug developers from all over the world are aware of the problem: in October 2017, the USA declared the opioid epidemic a national public health emergency [7], according to data showing that opioids are the eighth most common cause of premature death [2].

Opioids are widely indicated for pain management, anesthesia, and cough treatment [7,8,9]. At appropriate doses, patients are exposed to several side effects, such as constipation, nausea, sedation, hyperalgesia, sleep disorders, falls with increased rates of fractures in the elderly, etc. [10]. Most importantly, the use of opioids can cause drowsiness and sedation, which can affect the psychomotor performance and cognitive skills needed to drive safely: manual dexterity, hand‒eye coordination, mental alertness, and visual information processing are affected to different degrees [7,11]. Driving impairment is more prevalent during acute use or when the dosage is increased [8,12]. Thus, considering the risk of traffic collisions [8,13], opioid use should be monitored in the driver population. In Spain, the prevalence of opioid use among drivers is 1.8% and the use of multiple drugs is a common finding (56.2%), but an increase in consumption remains to be clarified [13].

This study, thus, presents the consumption of dispensed opioid analgesics by the population of Castile and Leon, Spain in 2015 to 2018. The length of use and particularly the use of opioid analgesics along with other driving impairing medicines (DIM) is also presented. We have considered as DIMs the medicines with the pictogram ‘medicines and driving’ on the packaging in Spain [14]. All analyses were done considering age and gender distributions, and the use of opioid analgesics alone or in combination with nonopioid analgesics. Adjusted consumption for licensed drivers is also presented in order to evaluate the differences in patterns of use of these medicines between the general population and drivers [14,15,16].

## 2. Experimental Section

This epidemiological, population-based registry study, presents the dispensation of opioid analgesics that belong to the Anatomical Therapeutic Chemical (ATC) classification N02A in 2015 to 2018. Information on such medicines and other DIMs dispensed to the population covered by the Spanish health system in Castile and León that is recorded in the CONCYLIA database (http://www.saludcastillayleon.es/portalmedicamento/es/indicadores-informes/concylia) was accessed (this information did not include medicines dispensed at hospitals or private clinics, or those considered ‘over the counter’ medications). As driving-impairing medicines (authorized medicines that can negatively affect fitness to drive or handle dangerous machinery), all opioid analgesics included the mandatory pictogram ‘medicines and driving’ on package leaflets and packs [17]. Appendix A presents all N02A opioids commercialized in Spain during the covered period according to their pharmaceutical formulation: (1) opioid alone (ATC codes N02AA, N02AB, N02AE, and N02AX), or (2) opioid in combination with a nonopioid drug (ATC codes N02AJ). Appendix A presents the age-distributed number of the population according to the CONCYLIA database and the adjusted proportion of drivers according to the Castile and León drivers’ license census data (http://www.dgt.es/es/seguridad-vial/estadisticas-e-indicadores/permisos-conduccion/) [14,15,16].

The findings presented here adhere to the standards of the reporting of studies conducted using observational routinely collected data (RECORD) statement [17]. This study was approved by the East Valladolid Health Area Ethics Committee on 17 March 2016 (reference number PI 16-387).

Any medicine dispensed following a physician prescription was considered medicine use. According to this consideration, the 2015 to 2018 dispensation (use) of opioid analgesics in Castile and León is presented as follows: (1) yearly frequency of N02A opioids consumption; (2) acute (1–7 days), subacute (8–29 days), and chronic use (≥ 30 days) of N02A opioids in each year; (3) yearly frequency of daily use of N02A opioids; and (4) concomitant use of opioid analgesics with other DIMs in each year. All analyses were done considering age and gender distributions.

Analyses gave us the frequencies (percentage) with their corresponding 95% confidence interval (95% CI) of opioid users, stratified by age and gender, and means accompanied by their standard deviations (SD) of opioid and DIM use, both in the general and the driver population. Differences between continuous variables (opioid and DIM use) were calculated using Student’s *t*-test (*t*), and those between categorical variables (opioid users) using Pearson’s Chi-squared test (*χ*^2^). The determination coefficient (*R*^2^) was calculated to measure the relationship between independent continuous variables. The level of significance was set at *p* ≤ 0.05. Statistical analysis was performed using the Statistical Package for the Social Sciences (SPSS version 24.0.; SPSS Inc, Chicago, IL, USA).

In addition, for the writing of the manuscript and the preparation of tables and figures, we used Microsoft Word and Excel (Microsoft Office version 365; Microsoft, Redmon, WA, USA).

## 3. Results

In total, 5,162,396 opioid drug packages were dispensed between 2015 and 2018 in Castile and León, Spain (2.4 million inhabitants), and 11.44% (274,563 individuals) of the general population took at least one analgesic opioid (see Appendix A for the age distribution of the general population and drivers). Overall, opioid use was associated with a mean ± SD of 2.54 ± 0.01 DIMs consumed, and was higher among women compared to men. Indeed, women consumed these medicines more frequently than men (13.73% versus 9.07%, *χ*^2^ = 3144.86, *p* = 0.001; Table 1), and this difference persisted among acute users (5.26%), chronic users (3.20%) and daily users (0.24%) of both types of formulations, with only one opioid and with a fixed combination of opioid analgesics and nonopioid analgesics (Table 1). With the exception of daily users, formulations containing paracetamol or a nonsteroidal anti-inflammatory drug were preferred (10.29% versus 1.78%, *χ*^2^ = 11,120.81, *p* = 0.001). In both sexes, consumption increased as age increased (Figure 1) for both types of formulations (Figure 2).

Most importantly, increases in the use of opioid analgesics among daily users (2.6 times higher) and chronic users (1.5% higher) were noted during the covered period (Figure 3), that supporting the overall increase in opioid use observed (1.5% higher), although the growths in percentage of the use of given formulations did not achieve significance (Table 2). The combinations codeine + paracetamol (N02AJ06, 6.09%) and tramadol + paracetamol (N02AJ13, 3.85%) represented more than three quarters of all opioid analgesic prescriptions. Tramadol (N02AX02, 0.98%) was the most common opioid analgesic prescribed alone. A preference for fentanyl (N02AJ13, 0.06%) was also noted among daily users. The following DIMs were used concomitantly by daily users: anxiolytics (N05B, 53.88%), antidepressants (N06A, 45.82%), other analgesics and antipyretics (N02B, 35.89%), antiepileptics (34.11%), hypnotics/sedatives (22.81%), and antipsychotics (12.02%). These trends in the concomitant use of other DIMs by daily users were similar among nondaily users.

Overall, among drivers, the use of opioid analgesics increased from 2015 to 2018. Nevertheless, figures differed among the driver population: opioid use (8.72%) was higher in men than in women (9.17% versus 8.06, *χ^2^* = 16,871.00, *p* = 0.001; Table 1). As with the general population, acute use (4.68%) predominates over chronic use (1.7%) and daily use (0.13%) among drivers. Interestingly, the higher percentage of male users in the driver population was not consistent with the DIM use that predominates among women into the general population (2.49 ± 0.02 versus 2.26 ± 0.01, *t* = −14.15, *p* = 0.001; Table 1). Importantly, opioid use increased with age, with a peak age range of 70–74 years in the case of male drivers and 55–59 years in the case of female drivers. (Figure 1). There were no other differences between drivers and the general population.

## 4. Discussion

In accordance with the data accessed and analyzed (5 million dispensations), the consumption of opioid analgesics was considerable and increased during the study period. Polydrug use was a common finding. Opioid analgesics were commonly prescribed for a short time period (1–7 days), although longer use (≥ 30 days) represented an important proportion. Women consumed more opioids than men, but drivers who took opioids were more frequently elderly men. The overall use increased as age increased. Formulations combining opioid analgesics with nonopioid analgesics were preferred. Risky populations should be the focus of all involved to help prevent driving under the influence.

Although the use of opioids in Spain may not be perceived as an “epidemic” like it is in the USA or Canada, the trends depicted here are alarming. Our figures are consistent with those obtained from other Western European populations, where increases of more than 4-fold were reached [3,4], showing that the risk of death due to opioid misuse stressed by U.S. authors is not as well recognized in Europe as it is in America [18,19]. Nevertheless, our analysis does not consider the nonmedical (recreational) use of opioids that is provided by other real-world studies [8]. According to administrative data on drug-positive roadside tests from 2011 to 2016, multidrug use involving a variety of illicit drugs in addition to illicit opioids is a fact that should be highlighted in our country [19]. Supporting this situation, our findings show that the consumption of other DIMs is common among opioid users. In any case, and only considering their medical use, the availability of new presentations for oral and nasal administration of opioids should also be mentioned as a cause for an increase in opioid use [20].

The common dosages suggest that physicians are following appropriate reasoning when prescribing these medicines for treating pain. Indeed, acute use was the most frequent situation, even among drivers. In addition, nonsignificant variations in using given formulations may be explained by the physician’s reasoning and habits of managing pain. Interestingly, irrespective of the pharmaceutical formulation, women more commonly used opioid analgesics, in most cases in association with other DIMs compared to men. This gender situation is consistent with the results if other studies covering European and Australian user populations [5,21,22], even if the age peaks for use may vary [23]. Furthermore, the use of opioids increased as age increased, which is easily understood because the disease burden increases as age increases. It should be taken into account that the misuse of psychoactive substances (mostly illegally obtained) is prevalent among young male users [24]. This fact may be the explanation for the differences between Europe and America concerning death due to opioid misuse, which is particularly common among middle-aged individuals in America [6]. This fact also highlights the risks of multidrug use among the elderly: driving impairment can lead to death or injuries in a traffic crash.

This study provides real-world evidence that contributes to the characterization of opioid consumers in Spain. Castile and Leon is the largest region of the country and previous analyses of opioid prescriptions have adequately reflected the situation at the national level [17], particularly given the objective to educate physicians and other healthcare professionals managing the patients who use these medicines. Notwithstanding, the limitations of any observational study must be highlighted. Registries and other real-world data sources should be considered as ‘emerging sources’ especially affected by confounders [25]. Importantly, all prescriptions’ information sources were not covered, which is probably the most obvious study limitation: selection bias should, thus, be taken into account as opioids dispensed at hospitals and dispensations obtained from private clinics were not included in our analyses (in Spain all opioids for medical use (ATC N02) must be by prescription, so there are no ‘over the counter’ dispensations). Finally, this study presents the adjusted proportions of drivers who are opioid consumers, but not an authentical stratification of use because the CONCYLIA database does not record information on driving [14,15,16].

## 5. Conclusions

In conclusion, from 2015 to 2018, the use of opioids increased in Spain. The concomitant use with other driving-impairing medicines was also important, and especially affecting women and the elderly. While the psychomotor-impairing effects of opioid analgesics are less marked in chronic use than in acute use, this study emphasizes the gravity of the situation in our country: frequent use of opioid analgesics with other DIMs may lead to potentially severe impairment of driving and an increase in accident risk. Furthermore, the acute and chronic use of these medicines was frequent. The combination of opioid analgesics with nonopioid analgesics was preferred. This study was intended to provide information and guidance for all those involved in managing patients’ use of opioids, including physicians and other healthcare providers, the authorities, patients and their families, drug developers, etc. Analyses of real-world sources may be challenging as the whole picture must be presented. Such analyses should, thus, be promoted to ensure clear responses to the problem of the consumption of driving-impairing medicines. Finally, the following (pragmatic) initiatives affecting the pharmacist‒physician relationship must be considered: providing clear messages on posology and the administration of opioids to the patients, the identification of misuse cases (e.g., by software improvements or direct pharmacist interventions), and developing strategies for the avoidance of unnecessary DIM use (e.g., via discussion panels, conjoint sessions, etc.). Opioid abuse represents a severe health burden all over the world, and could get worse in certain regions. Substance abuse habits in Spain should be taken seriously lest the situation worsen. The results of our study are also relevant to daily clinical practice because the appropriate use of opioids is a fundamental aspect of modern analgesia (physicians must prioritize nonopioid pain management strategies and reserve opioids for unmanageable/untreatable pain), and pain assessment is a fundamental issue in therapeutic planning (physicians must limit opioid use and the combined use with other DIMs, particularly in the elderly, with the objectives being to treat pain exclusively and not overtreat other medical/psychological conditions).

## Figures and Tables

**Figure 1 jcm-08-02148-f001:**
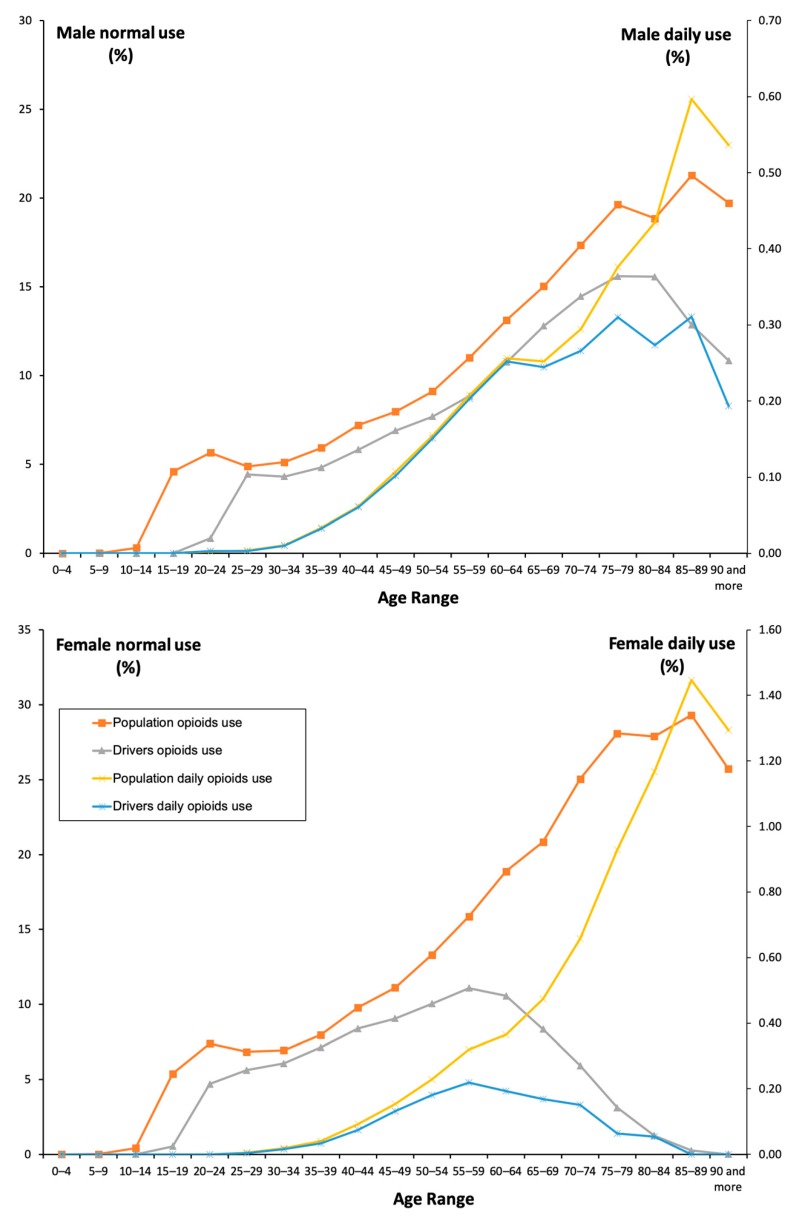
Frequency of opioid use by the general population and the driver population.

**Figure 2 jcm-08-02148-f002:**
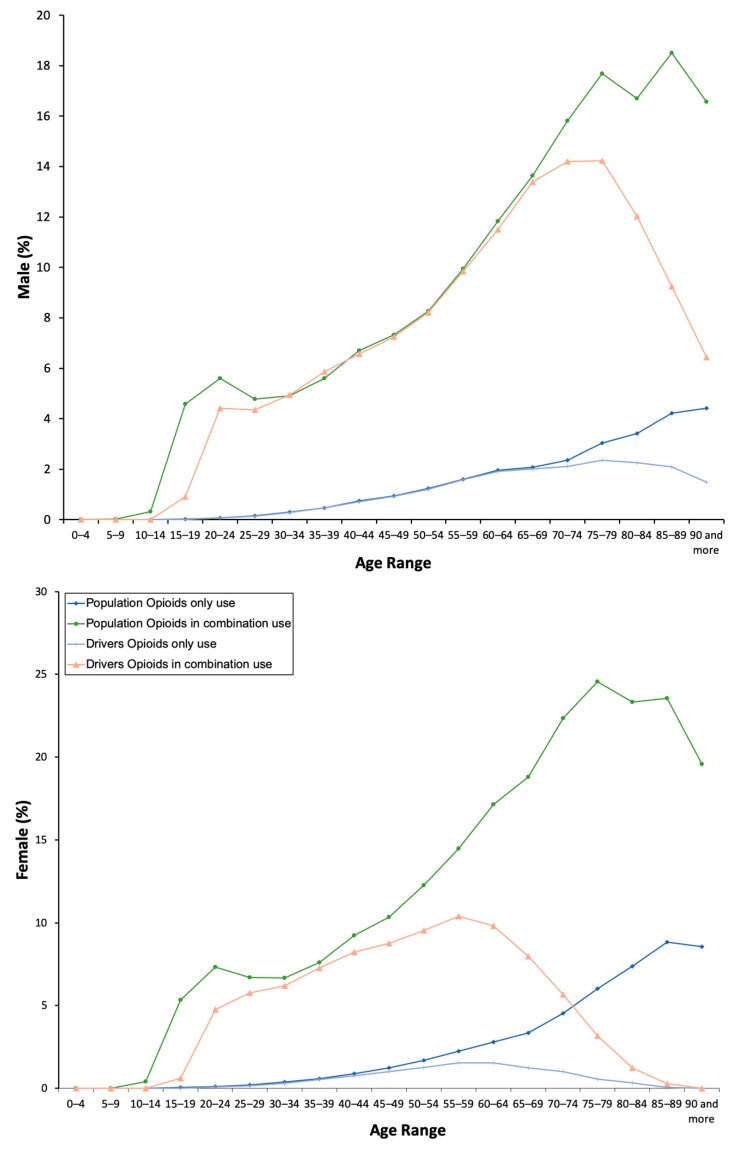
Frequency of opioid use as a “single drug” and “in combination.”

**Figure 3 jcm-08-02148-f003:**
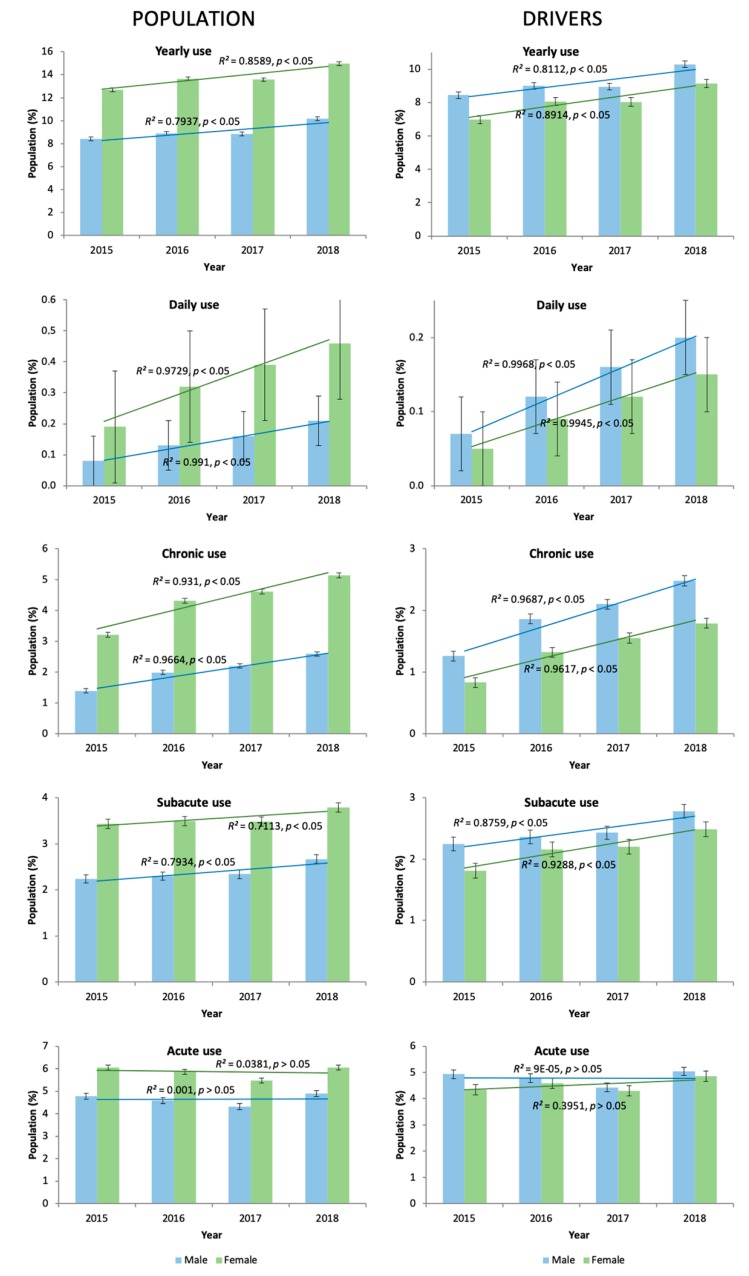
The evolution of opioid use in Castile and Leon (2015‒2018).

**Table 1 jcm-08-02148-t001:** Opioid consumption according to the CONCYLIA database and the Castile and León drivers’ license census data.

	Population Using Opioids % (95 CI)			Drivers Using Opioids % (95 CI)		
	Total	Opioids only	Opioids in Association	Total	Opioids only	Opioids in Association
Total	11.44 (11.32–11.56)	1.78 (1.66–1.91)	10.29 (10.17–10.41)	8.72 (8.57–8.88)	1.04 (0.88–1.2)	8.3 (8.14–8.45)
Male	9.07 (8.9–9.24)	1.19 (1.01–1.37)	8.27 (8.1–8.45)	9.17 (8.97–9.37)	1.17 (0.96–1.38)	8.59 (8.39–8.8)
Female	13.73 (13.56–13.89)	2.35 (2.18–2.53)	12.24 (12.07–12.41)	8.06 (7.81–8.3)	0.84 (0.59–1.1)	7.85 (7.6–8.1)
*χ*^2^, *p*	3144.861, *p* = 0.001	1570.787, *p* = 0.001	2003.986, *p* = 0.001	16,871.0, *p* = 0.001	1979.537, *p* = 0.001	15,501.654, *p* = 0.001
Type of use						
Chronic						
Total	3.2 (3.13–3.26)	1.03 (0.93–1.12)	2.69 (2.62–2.75)	1.7 (1.63–1.77)	0.51 (0.4–0.63)	2.15 (2.07–2.23)
Male	2.04 (1.95–2.12)	0.63 (0.5–0.76)	2.14 (2.05–2.23)	1.92 (1.83–2.02)	0.6 (0.45–0.74)	2.03 (1.93–2.14)
Female	4.32 (4.22–4.42)	1.41 (1.27–1.55)	3.21 (3.12–3.3)	1.37 (1.27–1.48)	0.39 (0.22–0.56)	2.07 (1.94–2.2)
*χ*^2^, *p*	1985.911, *p* = 0.001	1170.328, *p* = 0.001	1085.825, *p* = 0.001	4601.091, *p* = 0.001	1108.628, *p* = 0.001	3676.290, *p* = 0.001
Subacute						
Total	2.98 (2.91–3.04)	0.58 (0.5–0.65)	2.31 (2.25–2.37)	2.34 (2.26–2.42)	0.4 (0.3–0.5)	1.33 (1.27–1.4)
Male	2.39 (2.3–2.48)	0.43 (0.32–0.54)	1.48 (1.41–1.56)	2.46 (2.35–2.56)	0.44 (0.31–0.57)	1.17 (1.09–1.24)
Female	3.55 (3.46–3.63)	0.72 (0.62–0.82)	3.11 (3.02–3.2)	2.16 (2.03–2.29)	0.34 (0.18–0.5)	1.22 (1.12–1.32)
*χ*^2^, *p*	328.39, *p* = 0.001	293.255, *p* = 0.001	239.633, *p* = 0.001	4662.045, *p* = 0.001	614.929, *p* = 0.001	4406.436, *p* = 0.001
Acute						
Total	5.26 (5.18–5.35)	0.18 (0.14–0.22)	5.3 (5.21–5.38)	4.68 (4.57–4.8)	0.13 (0.07–0.19)	4.67 (4.55–4.79)
Male	4.65 (4.52–4.77)	0.13 (0.07–0.19)	4.65 (4.52–4.78)	4.79 (4.65–4.94)	0.14 (0.07–0.21)	4.62 (4.47–4.77)
Female	5.86 (5.75–5.98)	0.22 (0.17–0.28)	5.92 (5.8–6.04)	4.52 (4.33–4.71)	0.12 (0.02–0.21)	4.61 (4.41–4.8)
*χ*^2^, *p*	214.78. *p* = 0.001	76.966. *p* = 0.001	239.790. *p* = 0.001	7414.013. *p* = 0.001	258.198. *p* = 0.001	7365.078. *p* = 0.001
Daily use						
Total	0.24 (0.12–0.37)	0.13 (0.01–0.27)	0.11 (0.01–0.21)	0.13 (0.04–0.25)	0.07 (0.02–0.13)	0.08 (0.01–0.13)
Male	0.14 (0.01–0.30)	0.08 (0.01–0.17)	0.07 (0.01–0.13)	0.14 (0.07–0.28)	0.08 (0.03–0.14)	0.07 (0.01–0.15)
Female	0.34 (0.16–0.52)	0.18 (0.01–0.36)	0.16 (0.06–0.26)	0.1 (0.05–0.19)	0.05 (0.01–0.11)	0.05 (0.01–0.09)
*χ*^2^, *p*	317.198, *p* = 0.001	255.709, *p* = 0.001	266.803, *p* = 0.001	357.012, *p* = 0.001	160.226, *p* = 0.001	506.183, *p* = 0.00
Average of driving-impairing medicines; population opioid use						
Total	2.54 (2.53–2.54)	3.16 (3.14–3.18)	2.49 (2.49–2.50)	2.34 (2.33–2.36)	3.04 (3.01–3.07)	2.31 (2.30–2.33)
Male	2.28 (2.26–2.29)	2.87 (2.83–2.9)	2.24 (2.23–2.26)	2.26 (2.24–2.27)	2.86 (2.82–2.9)	2.22 (2.20–2.23)
Female	2.66 (2.65–2.67)	3.29 (3.27–3.32)	2.63 (2.61–2.64)	2.49 (2.47–2.51)	3.36 (3.3–3.42)	2.45 (2.43–2.47)
*t*, *p*	−30.520, *p* = 0.001	−19.564, *p* = 0.001	−41.237, *p* = 0.001	−14.153, *p* = 0.001	−14.425, *p* = 0.001	−16.290, *p* = 0.001
Average of driving-impairing medicines; population daily opioid use						
Total	3.52 (3.46–3.58)	3.77 (3.7–3.83)	3.23 (3.17–3.29)	3.67 (3.55–3.79)	3.93 (3.8–4.06)	3.31 (3.21–3.42)
Male	3.29 (3.16–3.41)	3.57 (3.44–3.69)	2.91 (2.8–3.02)	3.40 (3.25–3.54)	3.7 (3.55–3.85)	2.99 (2.87–3.12)
Female	3.60 (3.53–3.67)	3.86 (3.78–3.94)	3.35 (3.29–3.42)	4.18 (3.96–4.2)	4.41 (4.17–4.65)	3.82 (3.63–4.01)
*t*, *p*	−3.561, *p* = 0.012	−3.657, *p* = 0.001	−6.321, *p* = 0.001	−5.580, *p* = 0.001	−4.769, *p* = 0.001	−7.094, *p* = 0.001

Abbreviations: 95 CI. confidence interval.

**Table 2 jcm-08-02148-t002:** Evolution opioids consumption according to CONCYLIA database and the Castile and León drivers’ license census data (2015–2018).

	Population Using Opioids % (95 CI)				Drivers Using Opioids % (95 CI)			
	2015	2016	2017	2018	2015	2016	2017	2018
Total	10.59 (10.47–10.71)	11.31 (11.19–11.43)	11.25 (11.13–11.37)	12.6 (12.48–12.72)	7.86 (7.71–8.02)	8.63 (8.47–8.78)	8.59 (8.43–8.74)	9.82 (9.67–9.97)
Male	8.41 (8.23–8.58)	8.87 (8.7–9.05)	8.84 (8.67–9.02)	10.16 (9.99–10.33)	8.44 (8.24–8.64)	9 (8.8–9.2)	8.95 (8.75–9.15)	10.29 (10.09–10.49)
Female	12.7 (12.54–12.87)	13.66 (13.49–13.83)	13.57 (13.4–13.74)	14.97 (14.8–15.13)	6.99 (6.74–7.23)	8.07 (7.83–8.32)	8.04 (7.79–8.28)	9.13 (8.88–9.37)
*χ* ^2^ *, p*	3022.886, *p* = 0.001	2623.495, *p* = 0.001	2850.157, *p* = 0.001	3144.861, *p* = 0.001	15,215.257, *p* = 0.001	15,624.29, *p* = 0.001	15,862.716, *p* = 0.001	16,871, *p* = 0.001
Type of use								
Chronic								
Total	2.32 (2.26–2.38)	3.16 (3.1–3.23)	3.42 (3.35–3.49)	3.88 (3.81–3.96)	1.09 (1.03–1.15)	1.64 (1.57–1.71)	1.87 (1.8–1.95)	2.2 (2.12–2.27)
Male	1.39 (1.32–1.46)	1.98 (1.89–2.06)	2.19 (2.1–2.28)	2.59 (2.5–2.69)	1.26 (1.18–1.34)	1.86 (1.76–1.95)	2.1 (2–2.2)	2.48 (2.37–2.58)
Female	3.21 (3.13–3.3)	4.31 (4.21–4.41)	4.61 (4.51–4.71)	5.13 (5.03–5.24)	0.83 (0.75–0.92)	1.32 (1.22–1.43)	1.55 (1.44–1.66)	1.79 (1.68–1.9)
*χ* ^2^ *, p*	1558.149, *p* = 0.001	1617.673, *p* = 0.001	1907.393, *p* = 0.001	1985.911, *p* = 0.001	2548.811, *p* = 0.001	3635.145, *p* = 0.001	3924.455, *p* = 0.001	4601.091, *p* = 0.001
Subacute								
Total	2.84 (2.78––2.91)	2.91 (2.84–2.97)	2.92 (2.86–2.98)	3.24 (3.17–3.31)	2.07 (1.99–2.16)	2.28 (2.2––2.36)	2.34 (2.25–2.42)	2.66 (2.58–2.74)
Male	2.24 (2.15–2.33)	2.3 (2.21–2.4)	2.34 (2.25–2.43)	2.67 (2.58–2.77)	2.25 (2.14–2.35)	2.36 (2.26–2.47)	2.43 (2.32–2.54)	2.78 (2.67–2.89)
Female	3.43 (3.33–3.52)	3.49 (3.4–3.58)	3.48 (3.39–3.57)	3.79 (3.69–3.88)	1.81 (1.68–1.94)	2.16 (2.03–2.29)	2.2 (2.07–2.33)	2.49 (2.35–2.62)
*χ* ^2^ *, p*	434.257, *p* = 0.001	220.048, *p* = 0.001	254.568, *p* = 0.001	328.39, *p* = 0.001	4265.404, *p* = 0.001	4359.183, *p* = 0.001	4528.862, *p* = 0.001	4662.045, *p* = 0.001
Acute								
Total	5.43 (5.34–5.52)	5.24 (5.15–5.32)	4.91 (4.83–4.99)	5.48 (5.39–5.57)	4.7 (4.57–4.82)	4.7 (4.59–4.82)	4.37 (4.26–4.49)	4.96 (4.85–5.07)
Male	4.78 (4.65–4.91)	4.59 (4.47–4.72)	4.32 (4.19–4.44)	4.89 (4.76–5.02)	4.93 (4.77–5.08)	4.78 (4.63–4.93)	4.43 (4.28–4.57)	5.04 (4.89–5.18)
Female	6.06 (5.95–6.18)	5.86 (5.74–5.97)	5.48 (5.37–5.59)	6.05 (5.93–6.16)	4.34 (4.14–4.54)	4.59 (4.4–4.78)	4.3 (4.11–4.48)	4.85 (4.67–5.03)
*χ* ^2^ *, p*	313.76, *p* = 0.001	196.406, *p* = 0.001	208.622, *p* = 0.001	214.78, *p* = 0.001	8349.901, *p* = 0.001	7441.297, *p* = 0.001	7286.667, *p* = 0.001	7414.013, *p* = 0.001
Daily use								
Total	0.13 (0.01–0.26)	0.22 (0.1–0.35)	0.28 (0.15–0.41)	0.34 (0.21–0.46)	0.06 (0–0.11)	0.11 (0.01–0.21)	0.15 (0.05–0.25)	0.18 (0.02–0.34)
Male	0.08 (0–0.16)	0.13 (0.04–0.25)	0.16 (0.06–0.26)	0.21 (0.02–0.39)	0.07 (0.02–0.13)	0.12 (0.02–0.23)	0.16 (0.06–0.26)	0.2 (0.1–0.35)
Female	0.19 (0.01–0.36)	0.32 (0.14–0.49)	0.39 (0.21–0.57)	0.46 (0.28–0.64)	0.05 (0–0.1)	0.09 (0.04–0.16)	0.12 (0.02–0.23)	0.15 (0.05–0.25)
*χ* ^2^ *, p*	196.231, *p* = 0.001	256.032, *p* = 0.001	265.263, *p* = 0.001	317.198, *p* = 0.001	132.567, *p* = 0.001	214.021, *p* = 0.001	307.247, *p* = 0.001	357.012, *p* = 0.001
Average of driving-impairing medicines; population opioid use								
Total	2.51 (2.50–2.52)	2.55 (2.55–2.56)	2.53 (2.52–2.53)	2.55 (2.54–2.56)	2.30 (2.29–2.32)	2.35 (2.34–2.37)	2.36 (2.34–2.37)	2.38 (2.37–2.40)
Male	2.22 (2.21–2.24)	2.30 (2.28–2.31)	2.28 (2.26––2.29)	2.33 (2.32–2.34)	2.21 (2.10–2.22)	2.27 (2.25–2.28)	2.27 (2.25–2.28)	2.30 (2.29–2.32)
Female	2.65 (2.64–2.66)	2.68 (2.67–2.69)	2.65 (2.64–2.66)	2.67 (2.66–2.68)	2.47 (2.44–2.49)	2.49 (2.47–2.51)	2.50 (2.47–2.52)	2.51 (2.49–2.53)
*t, p*	−45.757, *p* = 0.001	−41.513, *p* = 0.001	−40.116, *p* = 0.001	−38.597, *p* = 0.001	−16.855, *p* = 0.001	−15.636, *p* = 0.001	−16.042, *p* = 0.001	−15.142, *p* = 0.001
Average of driving-impairing medicines; population daily opiois use								
Total	3.65 (3.57–3.73)	3.55 (3.49–3.61)	3.44 (3.39–3.50)	3.44 (3.34–3.49)	3.80 (3.63–3.87)	3.75 (3.63–3.87)	3.54 (3.43–3.65)	3.57 (3.47–3.67)
Male	3.42 (3.26–3.58)	3.42 (3.29–3.54)	3.15 (3.05––3.25)	3.17 (3.07–3.27)	3.53 (3.33–3.72)	3.51 (3.36–3.65)	3.27 (3.15–3.40)	3.27 (3.16–3.39)
Female	3.73 (3.64–3.83)	3.60 (3.53–3.67)	3.55 (3.49–3.62)	3.55 (3.48–3.61)	4.39 (4.05–4.73)	4.22 (4–4.24)	4.01 (3.81–4.21)	4.11 (3.92–4.29)
*t, p*	−3.263, *p* = 0.001	−2.553, *p* = 0.011	−6.410, *p* = 0.001	−6.443, *p* = 0.001	−4.579, *p* = 0.001	−5.230, *p* = 0.001	−6.110, *p* = 0.001	−7.451, *p* = 0.001

Abbreviations: 95 CI. confidence interval.

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
