# Peer review of "The Trends in Opioid Use in Castile and Leon, Spain: A Population-Based Registry Analysis of Dispensations in 2015 to 2018"

_jcm, 2019, doi:10.3390/jcm8122148_

Round 1

Reviewer 1 Report

The authors describe the increase in opioid prescribing in two regions in Spain, along with other driving impairing medicines and compared between the general population and the driving population.

The limitations of the study are addressed, most obvious that this does not include all prescribing sources and may not accurately reflect opioid use when those are taken into account. 

This is important as it draws light to an issue becoming increasingly problematic in the world, though not to the extent of Canada and the US, and allows an opportunity for recognition and intervention earlier.

Author Response

The authors describe the increase in opioid prescribing in two regions in Spain, along with other driving impairing medicines and compared between the general population and the driving population.

The limitations of the study are addressed, most obvious that this does not include all prescribing sources and may not accurately reflect opioid use when those are taken into account.

We thank the reviewer for these comments. Please, see at the last paragraph in Discussion section, lines 187 to 191, the text says now: “Importantly, all prescription drug information sources were not covered, which is probably the most obvious study limitation: selection bias should, thus, particularly be taken into account, as opioids dispensed at hospitals, and the dispensations in private clinics were not included in our analyses (in our country all opioids for medical use (ATC N02) must have mandatory prescription, so there were no ‘over the counter’ dispensations).”

This is important as it draws light to an issue becoming increasingly problematic in the world, though not to the extent of Canada and the US, and allows an opportunity for recognition and intervention earlier.

The reviewer is right and we thank once more his precious contribution. Please, see in conclusion section, lines 204 to 206, the text says now: “Analyses of real-world sources may be challenging as the totality of information should be presented. Such analyses should, thus, be promoted to ensure clear responses to avoid the problem of driving-impairing medicines consumption.”

Reviewer 2 Report

Congrats for the idea to study of opioid utilization in Spain. But, I think you may improve your work.

First at all, Castilla y León is a small region in Spain. You may describe gender, age in this region before to describe the results in the study.

of course the title may be changed by "in Castilla y Leon, Spain".

The introduction is too long. Every paragraph may show an idea: opioid use, secondary effects, traffic events. 

Methods: The design don´t explain correlation between opioids consum and traffic events. 

Results: So that, the results may be improve.

Such as conclusions and discusion, as well

Author Response

Congrats for the idea to study of opioid utilization in Spain. But, I think you may improve your work.

First at all, Castilla y León is a small region in Spain. You may describe gender, age in this region before to describe the results in the study.

We very thank the reviewer for these comments. Please, see at the last of first paragraph in in Experimental section, lines 85 and 86, the text says: “Supplementary Table 2 presents the population distribution by age (CONCYLIA database) and the adjusted proportion of drivers (drivers’ license census data ) in Castile and León, as previously made [17–19].”

Please, see also at beginning of the first paragraph in Results section, for a better understanding of percentage users calculation, the reader is now referred to this table available in our Supplementary material.

Please, find Supplementary material and the end of the manuscript file.

of course the title may be changed by "in Castilla y Leon, Spain".

Done. Please see the new title: “The trends in opioid use in Castile & Leon, Spain: A population-based registry analysis of the years 2015 to 2018 dispensation.”

The introduction is too long. Every paragraph may show an idea: opioid use, secondary effects, traffic events.

The reviewer is right and the text has been shortened. Now, Introduction section is composed of 3 paragraphs, and the second one explain in a concise manner secondary effects from opioids and their relation with driving risk concerns. Most importantly, the text now highlights the need to clarify an increase in the use of these medicines in a country where polydrug use is frequent (lines 62 to 64).

Methods: The design don´t explain correlation between opioids consumption and traffic events.

In our study, the correlation between opioid use and traffic events has not been measured. However, the method used to calculate the correlation for describing the trends into the percentage of the population that use opioids through the covered study period is presented in experimental section (lines 102 and 103).

Results: So that, the results may be improve.

Such as conclusions and discussion, as well

The reviewer is right. Critical details into Results section and explanations into Discussion section, that are highlighted, are added.

Round 2

Reviewer 2 Report

Would you please explain what is the clinical aplication to this investigation?

Author Response

Thank you for your consideration. The reviewer is right. Please, see at the end in Conclusion section, the following text has been added (lines 211 to 216):

“On the other hand, the results of our study are relevant in daily clinical practice: adequate use of opioids is a fundamental aspect in modern analgesia (physicians must prioritize non-opioid pain management strategies and to reserve opioids for unmanageable/untreatable pain), and pain assessment is a fundamental issue in therapeutic planning (physicians must limit opioid use and the combined use with other DIMs, particularly in the elderly, with the objectives to treat exclusively pain and not overtreat other medical/psychic conditions).”